# The procoagulant activity of tissue factor expressed on fibroblasts is increased by tissue factor-negative extracellular vesicles

**Marcela Rosas[1], David A. Slatter[1], Samya G. Obaji[1], Jason P. Webber[2], Jorge Alvarez-Jarreta[1], Christopher P. Thomas[3], Maceler Aldrovandi[1], Victoria J. Tyrrell[1], Peter V. Jenkins[1], Valerie B. O'Donnell[1], Peter W. Collins[1] ***

**1** Institute of Infection and Immunity, and Systems Immunity Research Institute, School of Medicine Cardiff University, Cardiff, United Kingdom, **2** Institute of Cancer and Genetics, School of Medicine, Cardiff University, Cardiff, United Kingdom, **3** School of Pharmacy and Pharmaceutical Sciences, Cardiff University, Cardiff, United Kingdom

* peter.collins@wales.nhs.uk

**Data Availability Statement:** All relevant data are available within the paper and its Supporting Information files.

## Abstract

Tissue factor (TF) is critical for the activation of blood coagulation. TF function is regulated by the amount of externalised phosphatidylserine (PS) and phosphatidylethanolamine (PE) on the surface of the cell in which it is expressed. We investigated the role PS and PE in fibroblast TF function. Fibroblasts expressed 6–9 x $10^4$ TF molecules/cell but had low specific activity for FXa generation. We confirmed that this was associated with minimal externalised PS and PE and characterised for the first time the molecular species of PS/PE demonstrating that these differed from those found in platelets. Mechanical damage of fibroblasts, used to simulate vascular injury, increased externalized PS/PE and led to a 7-fold increase in FXa generation that was inhibited by annexin V and an anti-TF antibody. Platelet-derived extracellular vesicles (EVs), that did not express TF, supported minimal FVIIa-dependent FXa generation but substantially increased fibroblast TF activity. This enhancement in fibroblast TF activity could also be achieved using synthetic liposomes comprising 10% PS without TF. In conclusion, despite high levels of surface TF expression, healthy fibroblasts express low levels of external-facing PS and PE limiting their ability to generate FXa. Addition of platelet-derived TF-negative EVs or artificial liposomes enhanced fibroblast TF activity in a PS dependent manner. These findings contribute information about the mechanisms that control TF function in the fibroblast membrane.

## Introduction

Tissue factor (TF) is a transmembrane protein that binds factor (F)VII/VIIa and acts as a cofactor for FVIIa-mediated cleavage of FX to FXa and FIX to FIXa initiating coagulation [1–3]. TF is required for normal haemostasis and is implicated in thrombotic disorders such as coronary artery disease and infection-induced disseminated intravascular coagulation [2–5]. TF is expressed constitutively on fibroblasts and smooth muscle cells that surround blood

**Funding:** Funding is acknowledged from: The British Heart Foundation (PG/14/29/30783) to PWC. Wellcome Trust (094143/Z/10/Z) to VOD European Research Council (LipidArrays) to VOD. CPT and VJT were funded by Medical Research Council grant (MR/M011445/1). The funders had no role in study design, data collection and analysis, decision to publish, or preparation of the manuscript.

**Competing interests:** No authors have competing interests.

vessels [6–8]. After vessel damage, circulating FVII/FVIIa interacts with TF expressed on the fibroblast membrane, initiating coagulation to arrest bleeding. In atherosclerotic lesions, TF is also associated with vascular smooth muscle cells, foam cells and monocytes contributing to pathological thrombosis at the time of plaque rupture [9, 10].

TF initiates coagulation on membrane surfaces. *In vitro* experiments using synthetic membranes demonstrate that phospholipids regulate TF activity [11]. Specifically, phosphatidylserine (PS) is required for TF function whilst other phospholipids, such as phosphatidylethanolamine (PE), enhance the action of PS [12]. In contrast, native phosphatidylcholine (PC) does not support coagulation [12]. We have shown that the fatty acyl chains of phospholipids affect coagulation, with enzymatically oxidised phospholipids supporting coagulation factor binding better than non-oxidized phospholipids *in vitro* and *in vivo* [13–15].

*In vitro*, TF activity is optimal when incorporated into membranes comprising 30% PS/PE [16, 17], however, whether this amount of PS/PE is available on the surface of TF-bearing cells is undetermined. The molecular species of PS/PE and the amount that are externally facing in a TF-bearing membrane has not been investigated. Cellular TF activity can be enhanced (decrypted) by mechanisms that include increasing externalized PS/PE, by processes such as apoptosis, cell activation and damage, and the action of protein disulphide-isomerise [6, 7, 18–20]. Following TF decryption, there is a 2–10 fold increase in FXa generation. Increased cell-bound TF activity has been demonstrated following cell disruption by freeze/thawing [4], apoptosis and treatment with 4-hydroxy 2-nonenal or $HgCl_2$ [21, 22].

It is often assumed that only PS in the TF-bearing membrane affects TF function, however, cell-derived extracellular vesicles (EVs) are another potential source of PS. EVs are shed from the plasma membrane of platelets, monocytes and endothelial cells and contribute to clot formation *in vivo* due to recruitment of TF-bearing EVs into a developing thrombus [23, 24]. In cancer, TF-positive EVs are incorporated into cells with low or no TF expression (e.g. endothelial cells), enhancing their pro-coagulant activity and the development of thrombosis [25]. TF is absent on EVs from patients with venous thrombosis [26, 27], suggesting that these vesicles may contribute to thrombotic processes via alternative mechanisms that are not dependent on TF. Here we characterise the the phospholipid environment of TF in fibroblasts and investigate the effect of platelet-derived EVs, which did not contain TF, and TF-negative PS-containing liposomes, as a source of phospholipid exogenous to the cell membrane, on TF function in fibroblasts.

## Methods

### Cells

As a model for human fibroblasts we used HCA2-hTERT cells, a gift of Professor D Aeshcilimann (Cardiff University), derived from neonatal normal diploid foreskin and immortalised with pBABE-hTERT to rescue cells from senescence by expression of the catalytic subunit of human telomerase [28]. HCA2 fibroblasts were cultured in DMEM with 10% fetal bovine serum (FBS). To induce mechanical damage, fibroblasts were scraped with a pipette tip, and then resuspended into single cells with pipetting, prior to further assessment of PS/PE externalisation and FXa generation.

### Reagents

Fibroblasts were lifted with Accumax™ cell detachment solution (Innovative Cell Technologies, San Diego, CA, USA). Annexin V (unlabelled, or labelled with Pacific Blue or Alexa 488), Annexin V Binding buffer, anti-human CD142 (clone NY2: Tissue Factor) and Mouse IgG-1k isotype antibodies (labelled with Allophycocyanin (APC), and Propidium Iodide (PI) Solution

were from BioLegend (San Diego, CA, USA). Purified mouse anti-human TF monoclonal antibody (clone HTF-1) and mouse IgG1 isotype control (MOPC-21) were purchased from BD Biosciences (BD Pharmingen™, San Diego, CA, USA). Quantum Simply Cellular anti-Mouse IgG beads were from Bangs Laboratories (Fishers, IN, USA). FVIIa was from Novonordisk (Bagsvaerd, Denmark), FX and FXa were from Enzyme Research Laboratories (Swansea, UK). Tyrode's buffer was homemade (134 mM NaCl, 12mM $NaHCO_3$, 2.9mM KCl, 0.34mM Na2HPO$_4$, 1mM $MgCl_2$, 10mM HEPES 5mM Glucose).

## Lipids

1,2-distearoyl-*sn*-glycero-3-phosphocholine (DSPC, PC(18:0_18:0)), 1-stearoyl-2-arachido-noyl-*sn*-glycero-3-phosphocholine (SAPC, PC(18:0_20:4)), 1-stearoyl-2-arachidonoyl-*sn*-glycero-3-phosphoethanolamine (SAPE, PE(18:0_20:4)) and 1-stearoyl-2-arachidonoyl-*sn*-glycero-3-phospho-L-serine (SAPS, PS(18:0_20:4)) were from Avanti® Polar Lipids, Inc (Alabaster, AL, USA). 1,2-dimyristoyl-sn-glycero3-PS (DMPS, PS(14:0_14:0)) and 1,2-dimyristoyl-sn-glycero-3-PE (DMPE, PE(14:0_14:0)) were biotinylated and used as internal standards to assess PS/PE externalization by mass spectrometry, using N-Hydroxysuccinimide (NHS)-biotin as described [15, 29].

## Liposomes

Liposomes were prepared as described [15]. Briefly, lipids were dried in a glass vial by evaporation under $N_2$ and suspended in 20 mM HEPES, 154 mM NaCl, pH 7.35 by vortexing. Liposomes were generated by 10 freeze-thaw cycles using warm water and liquid nitrogen, followed by passing through Liposofast mini-extruder with 100-nm pore polycarbonate membranes (Avestin, Otawa, Canada) 19 times. Liposomes of the following compositions were made: 100% PC (65% DSPC and 35% SAPC), PC/10% PS (65% DSPC, 25% SAPC and 10% SAPS) PC/10% PE (65% DSPC, 25% SAPC, 10% SAPE). Liposomes were added to $10^4$ fibroblasts or buffer and incubated for 10 min at 37˚C/5% $CO_2$. For binding experiments, liposomes contained 1%SAPE-biotin, generated by reacting N-hydroxysuccinimide-biotin with PE followed by HPLC purification [14].

## Platelet-derived extracellular vesicles

Platelets were isolated from healthy volunteers recruited under a protocol approved by the School of Medicine Research Ethics Committee at Cardiff University (reference 16/02). Blood was taken into acid citrate dextrose (ACD) and centrifuged at 250g for 10min (brake off) to obtain platelet-rich plasma (PRP). Platelets were pelleted at 900g for 10min. Platelets were resuspended in Tyrode's buffer at $2x10^8$ /ml. Platelets were pelleted at 900g for 10min. All the centrifugation was at room temperature. Platelets were resuspended in Tyrode's buffer at $2x10^8$ platelets/ml. Platelets were either untreated (UT) or stimulated with 10 μM $Ca^{++}$ ionophore A23187 (Sigma, Saint Louis, MO, USA) or 0.5% DMSO (vehicle control) for 30 min at room temperature. Samples were spun down at 1000g for 15 min twice to remove any residual platelets, and EVs were obtained by centrifugation at 16000 g for 30 min. To determine the ability of EVs to support FXa generation, 40 μl was added to $10^4$ fibroblasts or EVs were assayed in absence of cells for 10 min at 37˚C and FVIIa-dependent FXa generation measured as below.

The size and number of EVs was determined by Nanoparticle Tracking Analysis (NTA) using a NanoSight NS3000 (Malvern P Malvern Panalytical Ltd, Malvern, UK) with a 488 nm laser and temperature set to 25˚C. Six videos of 60 s were taken in light scatter mode with controlled fluid flow with a pump speed set to 80. Videos were analysed using the batch analysis

tool of NTA 3.1 software (version 3.1 build 3.1.54), where minimum particle size, track length and blur were set at "automatic". The area under the histogram for each triplicate measurement was averaged and used as a particle concentration measurement. Background measurements of culture media that had not been exposed to cells contained negligible particles. Data were collected and averaged from three independent donors.

## Assessment of fibroblast TF expression

Quantification of molecules on the surface of the cells using flow cytometry has been reported previously [30]. A standard curve was created based on the intensity of fluorescence given by the mouse monoclonal TF antibody (clone NY2 labelled with Allophycocyanin; Biolegend, San Diego, CA, USA) when it bound microbeads containing increasing concentrations of anti-mouse IgG (Quantum™ Simply Cellular®). The intensity of fluorescence was proportional to the number of IgG antibody molecules on each bead population, and these values were converted to number of TF molecules/cell using a QuickCal® spreadsheet (Bangs Laboratories, Fisher, IN, USA). The background binding of an isotype control was subtracted.

## Externalisation of PS/PE by Annexin V binding

Cells were incubated with annexin V-Pacific Blue for 15 min and 1 µg/ml, propidium iodide (PI) was used to exclude dead cells. Percentage of cells binding annexin V was assessed by flow cytometry using a Cyan ADP (Beckman Coulter, Brea, CA, USA). For microscopy, cells were incubated with annexin V-A488 and analysed with MetaMorph® imaging software (Molecular Devices, San Jose, CA, USA).

## FXa generation assay

Fibroblasts were cultured in flat-bottom 96-well plates at $10^4$ cells/well. After washing with PBS FVIIa (10nM) and FX (136nM) were added. Z-Gly-Gly-Arg-AMC (Bachem, Bubendorf, Switzerland), a substrate for FXa [31], was resuspended in 20mM HEPES pH 7.35, 0.2% sodium azide, 0.5% BSA, and 100mM $CaCl_2$ and dispensed into the plate (20 µl). Substrate cleavage was determined by fluorescence using a Fluroskan Ascent™ microplate fluorometer (ThermoFisher, Helsinki, Finland).

## Measurement of PS and PE by mass spectrometry

Lipids were extracted from fibroblasts ($10^6$) using a modified Bligh and Dyer method [13, 29]. Samples (400 µL) were added to 1.5 mL of chloroform and methanol (1:2) and vortexed. A further 0.5 mL chloroform was added, followed by 0.5 mL water with vortexing. Samples were centrifuged for 5 min at 500g. The lower chloroform phase was recovered, dried, dissolved in methanol, and stored at −80˚C [13, 29].

Phospholipid precursor scans were run using a QTRAP 6500 LC-MS/MS system (Sciex, Framingham, MA, USA) to determine the composition of PS and PE species. Briefly, lipid extracts were separated by reverse phase HPLC, in negative ion mode, using a Luna 3µm C18 150 × 2-mm column (Phenomenex, Torrance, CA). Lipids were separated using gradient chromatography of 50–100% solvent B over 10 minutes followed by 30 minutes at 100% solvent B with a flow rate of 200ul/min. (Solvent A: methanol:acetonitrile:water, 1mM Ammonium Acetate, 60:20:20. Solvent B: methanol, 1mM Ammonium Acetate). Lipid extracts were analysed for PS by neutral loss of 87 atomic mass units (amu) in negative mode. Spectra were acquired scanning Q1 from 700–870 amu with total scan time (including pauses) over 1.5 s. For PE, spectra were acquired scanning Q1 from 650–850 amu with total scan time over 1.7 s, with

precursor ion scanning at *m/z* 196. Mass spectrometer parameters were as follows: Curtain gas: 35, IS -4500, TEM 500˚C, GS1 40, GS2 30, DP -100, CE -30.

To quantify the amount and percentage of externalised PS and PE, fibroblasts ($10^6$) were incubated with either EZ-Link™ Sulfo-NHS-Biotin in PBS to measure PS/PE externalisation or EZ-Link™ NHS-Biotin (Pierce Biotechnology, Rockford, lL, USA) in DMSO to determine total PS/PE. In this assay, biotin is bound to the external-facing amino groups present on PS and PE but does not bind to other PLs such as PC. Lipid extraction was performed as above, and any biotinylated proteins labelled on the cell surface were removed in the upper aqueous phase. The externally facing PLs were characterized by mass spectrometry. Biotinylated PLs were separated on an Ascentis C18 column (5 μm, 150 × 2.1 mm, Sigma-Aldrich) using isocratic flow of mobile phase methanol with 0.2% (wt/vol) ammonium acetate at 0.4ml/min for 27 minutes. Biotinlylated PLs were identified using MRM runs in negative mode. Mass spectrometer conditions were as follows: curtain gas 30, IS-4500, TEM 550˚C, GS1 40, GS2 30. Declustering potential (DP) ranged from -135 to -170, and collision energy (CE) ranged from -42 to -60. Peaks were integrated using Analyst 1.7 software (AB Sciex™, CA, US), and the ratio of the analyte to the internal standard was used to control for losses during extraction. Lipids were quantified using a standard curve. Inclusion criteria for chromatographic peaks were those which exceed a signal:noise ratio of 3 and had 7 or more points across the peak [13, 29].

### TF-phospholipid binding assay

Polysorp plates (Nunc®, Rochester, NY, USA) were coated overnight with 5 μg/ml NeutrAvidin protein (Pierce Biotechnology, Rockford, lL, USA). Next day, plates were washed with Annexin V Binding Buffer (ABB) and blocked with 0.5% fat-free BSA in PBS. Biotinylated liposomes were captured at room temperature for 2 hours. Alternatively, phospholipids (DSPC, SAPC, SAPE, and SAPS) were diluted in methanol at 3 μg/ml and air-dried on Polysorp plates for 90 min at room temperature. Blocking was with 0.5% gelatin (Sigma, Saint Louis, MO, USA). For both assays, the extracellular domain (G2-E219) of tissue factor (R&D Systems Minneapolis, MN, USA) was added at 1 μg/ml in 10mM HEPES pH7.4, 140mM NaCl, 2.5mM $CaCl_2$ and incubated with either liposomes or dried lipid for 2 hours at room temperature. Purified anti-human TF antibody was incubated for 1 hour. After washing, antimouse IgG-peroxidase (Sigma, Saint Louis, MO, USA) was added. Lastly, plates were washed again with ABB and SuperSignal® ELISA Pico Chemiluminescent Substrate (Pierce Biotechnology, Rockford, lL, USA) was used as substrate. Graphs show relative luminescence units (RLU).

### Statistical analysis

Statistical analysis was performed using GraphPad Prism version 7.00 for Windows (GraphPad Software, La Jolla, CA, USA). Data are presented as median, 25th to 75th centiles and ranges or mean ± standard error of the mean (SEM). Comparisons use the Mann-Whitney U test.

## Results

### Characterisation of phospholipid species in platelets and fibroblasts

The phospholipid environment of TF in the fibroblast cell membrane has not been described in detail. We compared the molecular species of PS and PE in platelets and fibroblasts (Fig 1). Platelets contain two prominent species of PS: PS (36:1), *m/z* 788.6 (1-stearoyl-2-oleoyl-phosphatidylserine, SOPS) and PS (38:4), *m/z* 810.7 (1-stearoyl-2-arachidonoyl-phosphoserine,

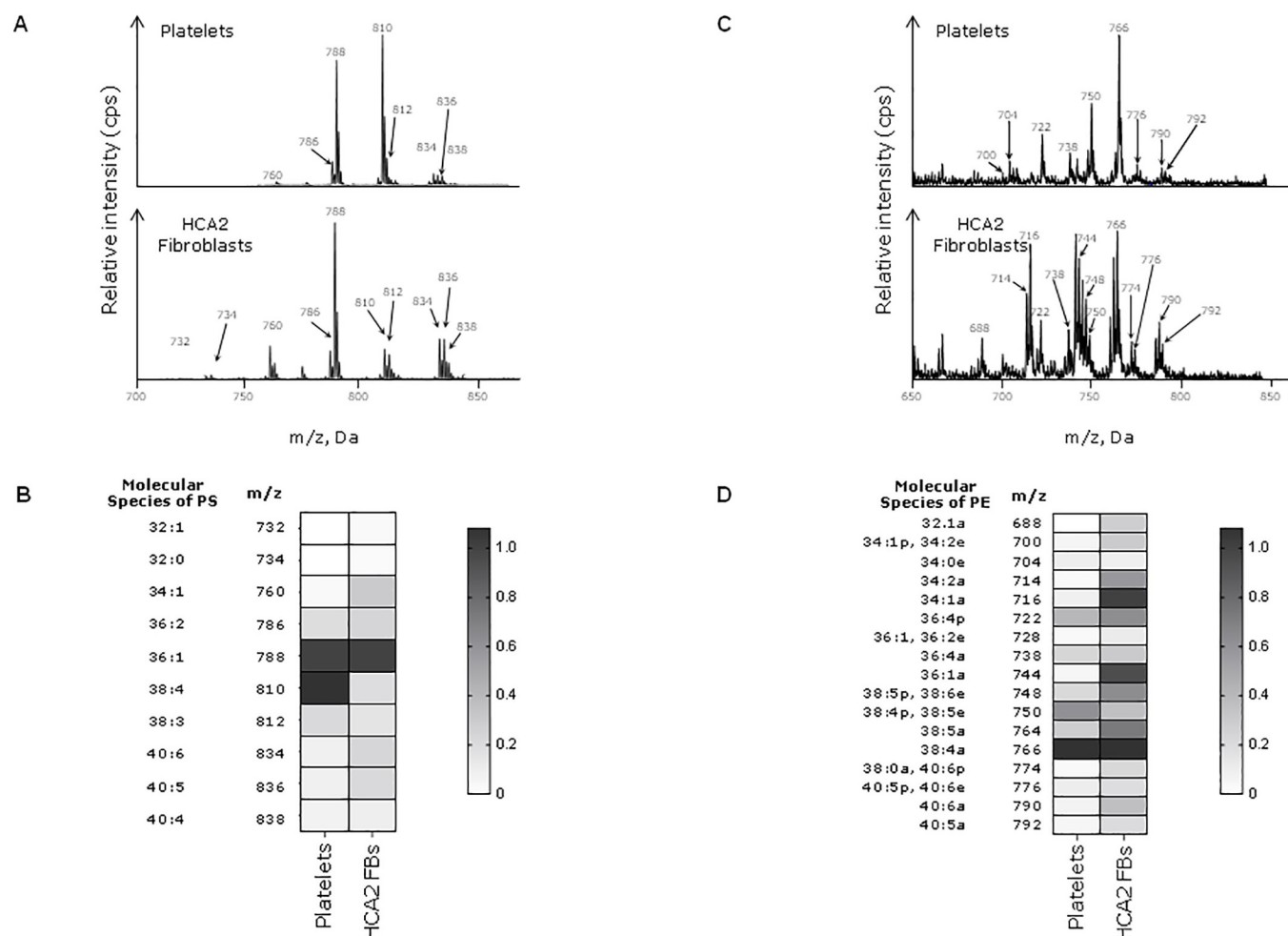

**Fig 1. PS/PE species in platelets and fibroblasts.** Lipids were extracted from $10^6$ unstimulated fibroblasts and 2 x $10^8$ freshly-isolated human platelets. PS/PE precursor scans were run using a QTRAP 6500 LC-MS/MS system as specified in materials and methods. Data are representative of three independent experiments. **A)** Mass spectra of PS species showing the mass-to-charge ratio (m/z) and the relative intensity (cps) of each species. **B)** PS intensity values from the spectra represented on the Heatmap. **C)** Mass spectra of PE species. **D)** Heat map representing PE intensity values. Molecular species refers to the total number of carbon atoms and double bonds in the two fatty acid chains of the phospholipid.

SAPS) [13, 29]. In contrast, SOPS alone is the main species in fibroblasts (Fig 1A and 1B). The main molecular species of PE found in platelets is PE (38:4), *m/z* 766 (1-stearoyl-2-arachido-noyl-sn-glycero-3-PE, SAPE) [13, 29], which is also found in fibroblasts. However, other PE species are found in fibroblasts including PE (36:1a), *m/z* 744, and PE (34:1a), *m/z* 716 (Fig 1C and 1D). These findings demonstrate that fibroblasts have different molecular species of phospholipids capable of supporting coagulation compared to platelets. However, it is the externally facing PS and PE in the TF-bearing membrane that are important for control of haemostasis and this was investigated next.

### Adherent fibroblasts have low amounts of external facing PS and PE which increase after cell damage

The amount of PS and PE in the external leaflet of the fibroblast cell membrane in unperturbed and mechanically damaged fibroblasts has not been quantified previously. Flow cytometry showed that only 2% of live fibroblasts were annexin V positive when cells were lifted gently

with Accumax (Fig 2A upper panel). The very low number of positive cells was confirmed visually by fluorescent microscopy with adherent fibroblasts (Fig 2B, upper panel). Fibroblasts undergoing staurosporine-induced apoptosis, which leads to externalisation of PS/PE, were used as a positive control (Fig 2B, lower panel). Annexin V binding does not identify or quantify the specific molecular species of externally facing PS and PE and this was addressed by mass spectrometry. In unperturbed adherent fibroblasts about 1% of total cellular PS and PE were found on the outer leaflet of the plasma membrane (Fig 2C). The amount of externally-facing PS and PE on adherent, unperturbed fibroblasts is shown in Table 1. These low levels of

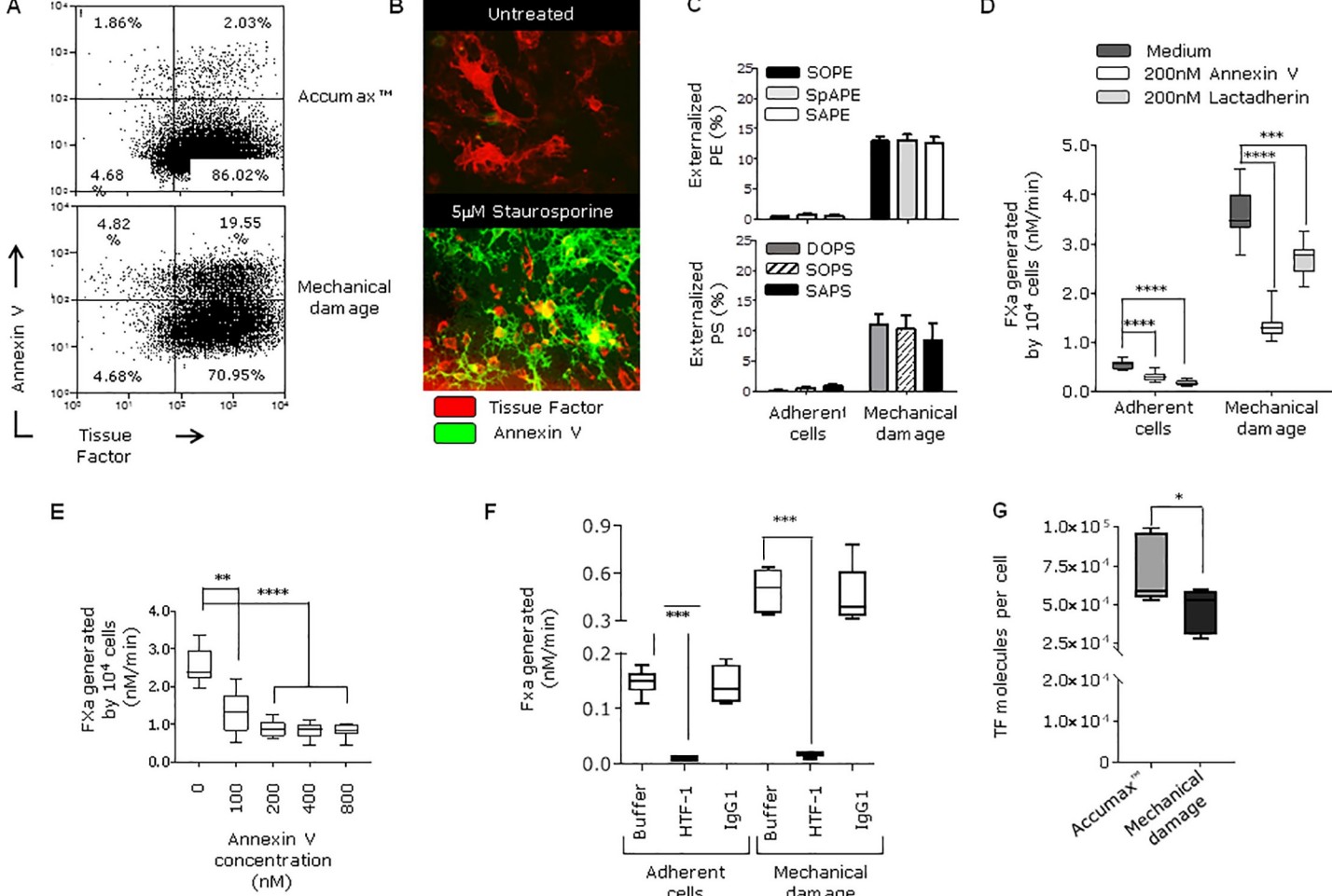

**Fig 2. Externalization of PS/PE, expression of TF and pro-coagulant activity on healthy and damaged fibroblasts. A)** Flow cytometry analysis of HCA2 fibroblasts lifted either using Accumax™ cell detachment solution to avoid membrane disruption or by scraping to induce mechanical damage. Cells were stained with Annexin V-Pacific Blue and anti-TF antibody labelled with APC. Plots show cell populations after propidium iodine was used to exclude dead cells. **B)** Adherent HCA2 fibroblasts were cultured either with medium alone (untreated) or with 5μM staurosporine for 1 hour at 37°C to induce membrane damage due to apoptosis (positive control). Cells were stained with TF antibody (NY2) labelled with APC and Annexin V-A488 for analysis by microscopy. **C)** Percentage of total and externalized PS and PE species was determined by MS (mean ± SEM) as indicated in materials and methods. Amounts (ng) of externalized PS and PE are presented in Table 1. **D)** FXa generation was measured on adherent cells and cells after mechanical damage. The effect of PS-binding proteins was assessed by pre-incubating the cells with 200 nM Annexin V or Lactadherin for 30 min at room temperature before adding coagulation factors (FVIIa and FX) ****p <0.0001, ***p <0.001 **E)** Mechanically-damaged fibroblasts were pre-incubated with increasing concentrations of annexin V prior the addition of FVIIa and FX, ****p <0.0001, **p = 0.005. **F)** FXa generation by fibroblasts and was measured when cells were adhered or after mechanical damage. TF antibody (HTF-1) or isotype control (10 μg/ml) were pre-incubated with cells before adding FVIIa/FX, and FXa production was assessed, ***p <0.001. **G)** The number of TF molecules on HCA2 fibroblasts was determined by flow cytometry as described in materials and methods. Cells were labelled with TF antibody-APC after Accumax treatment or scraping. Experiments in this figure were repeated two or three times.

**Table 1. Quantification of externally facing phosphatidylserine and phosphatidylethanolamine on fibroblasts before and after mechanical damage.**

| | | Adherent fibroblasts | Mechanically damaged fibroblasts | Fold increase |
|---|---|---|---|---|
| | | Mean (SE) ng/10⁶ cells | | |
| **Phosphatidylserine** | 1-Stearoyl-2-oleoyl-phosphatidylserine (SOPS) | 1.01 (0.08) | 14.29 (2.05) | 14.1 |
| | 1-Stearoyl-2-arachidonyl-phosphatidylserine (SAPS) | 0.09 (0.02) | 0.61 (0.11) | 6.8 |
| | 1,2-Dioleoyl-phosphoserine (DOPS) | 0.05 (0.03) | 3.18 (0.40) | 63.6 |
| **Phosphatidylethanolamine** | 1-Stearoyl-2-arachidonyl-phosphatidylethanolamine (SAPE) | 1.30 (0.34) | 18.41 (1.43) | 14.2 |
| | 1-stearoyl-2-arachidonoyl-sn-glycero-3-phosphoethanolamine (SpAPE) | 0.64 (0.06) | 7.06 (0.52) | 11.0 |
| | 1-Stearoyl-2-oleoyl-phosphatidylethanolamine (SOPE) | 0.56 (0.14) | 11.22 (0.63) | 20.0 |

The amount of the predominant molecular species of phosphatidylserine and phosphatidylethanolamine, present on the external surface of the fibroblast plasma membrane, was quantified by mass spectrometry. Very low levels were seen in adherent unperturbed fibroblasts but this was increased after mechanical damage used to simulate tissue injury.

externalised PS/PE contributed to the small amount of TF activity found on adherent, unperturbed fibroblasts because the limited amount of FVIIa-dependent FXa generation was decreased by incubation with annexin V or lactadherin (Fig 2D and 2E) and almost entirely inhibited by an anti-TF antibody (Fig 2F).

To simulate tissue damage at the time of vessel injury, fibroblasts were scraped from the culture flask (mechanical damage). The number of TF molecules/cell on fibroblasts and the variation in TF expression between individual cells was assessed by flow cytometry (Fig 2G). Mechanical damage led to a 7-fold increase in TF/FVIIa-dependent FXa generation on fibroblasts (Fig 2D). The increased FXa generation was reduced by 65% when pre-incubated with 200 nM annexin V and 25% when pre-incubated with lactadherin, confirming that externalised PS and or PE were influencing the enhanced TF function (Fig 2D and 2E). The enhanced FXa generation was almost completely inhibited by an anti-TF antibody (Fig 2F). Following mechanical damage the proportion of live fibroblasts binding annexin V increased from 2% to 19.6% (Fig 2A, lower panels) and the proportion of PS and PE that was on the external surface of the membrane increased to about 13% of total PS/PE (Fig 2C). The amount of externally-facing PS increased 7- to 64-fold, dependent on the molecular species, and PE increased 11- to 20-fold (Table 1). SOPS and SAPE were the most abundant externally-facing aminophospholipids (Table 1).

## Platelet-derived extracellular vesicles and PS-containing liposomes enhance cellular TF function

The previous experiments show that the amount of externalized PS and PE on unperturbed fibroblasts is low and TF/FVIIa-dependent FXa generation limited. Since TF/FVIIa activity is required for normal haemostasis, and controlled by PS exposure, we asked whether TF function could be enhanced by a source of TF-negative PS that was exogenous to the TF-bearing cell.

To test this, PS-expressing EVs from the supernatant of unstimulated platelets and EVs generated from platelets treated with Ca$^{++}$ ionophore A23187 were used [32]. The number and size of the EVs was determined using NTA (Fig 3A). The median (25th to 75th centiles) size of the generated EVs was 209 (203.8 to 225.7) nm. MS analysis revealed that the platelet-derived EVs expressed high levels of PS and PE species on their external leaflet as shown on Fig 3B and 3C. When EVs from unstimulated platelets were added to 10⁴ fibroblasts they enhanced

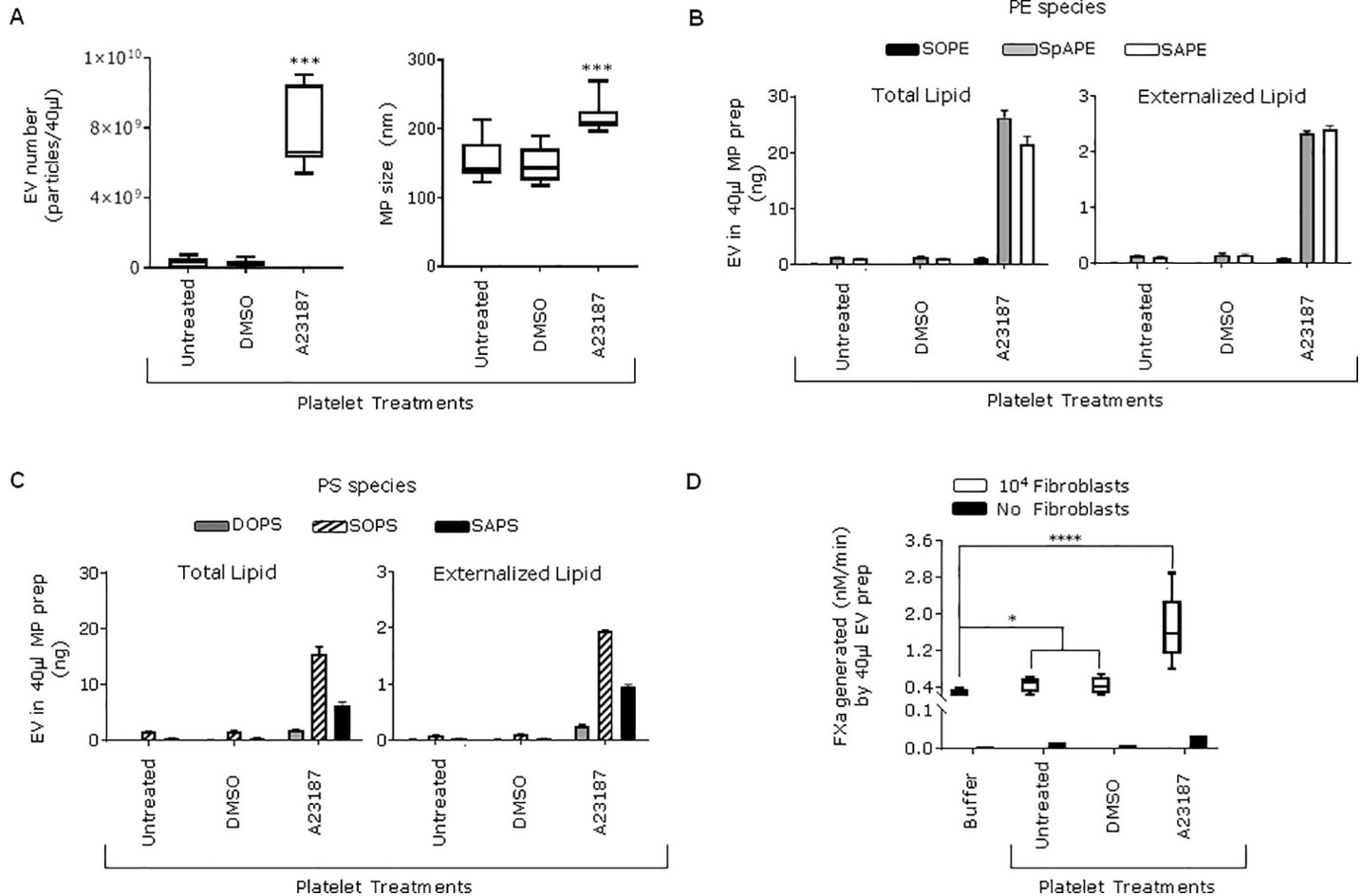

**Fig 3. Addition of platelet-derived extracellular vesicles enhances the procoagulant activity of fibroblasts.** Platelet-derived EVs were isolated from unstimulated platelet supernatant (untreated) or treated with the ionophore A23187. DMSO is the vehicle control for ionophore. **A)** The number and size of EVs was determined using Nanoparticle Tracking Analysis as described in materials and methods. **B)** Quantification of total and externalized PE and **C)** PS on EVs was performed by MS (mean ± SEM). **D)** EVs were added to either $10^4$ fibroblasts or in absence of cells for 10 min at 37°C and FXa generation measured as described in materials and methods. Fibroblasts or no cells were also loaded with only buffer but not platelet-derived treatments, ****p <0.0001, *p <0.05. Data correspond to EVs generated from three independent donors. PS/PE species were quantified on EVs derived from one of those donors.

FVIIa-dependent FXa generation 2-fold whilst when ionophore–generated EVs were used they enhanced FXa generation 6-fold. EVs derived from unstimulated and ionophore activated platelets supporting almost no FXa production in the absence of fibroblasts, indicating that the platelet-derived EVs had no TF activity (Fig 3D).

To examine the contribution of PS and or PE to the enhanced FXa generation seen when platelet-derived EVs were added to fibroblasts, we compared the effect of adding synthetic 100% PC liposomes with those containing 10% PS and 90% PC or 10 PE and 90% PC. None of these liposomes contained TF and, as expected, did not support FVIIa-dependent FXa production (Fig 4A). However, liposomes containing 10% PS enhanced fibroblast FXa generation in a concentration dependent manner (Fig 4A) achieving similar levels to those observed when activated-platelet EVs were added. Neither 100% PC (as expected) nor 10% PE liposomes enhanced FXa generation on fibroblasts (Fig 4A). The enhanced FVIIa-dependent FXa generation seen when PS containing liposomes were added to fibroblasts was inhibited by annexin V

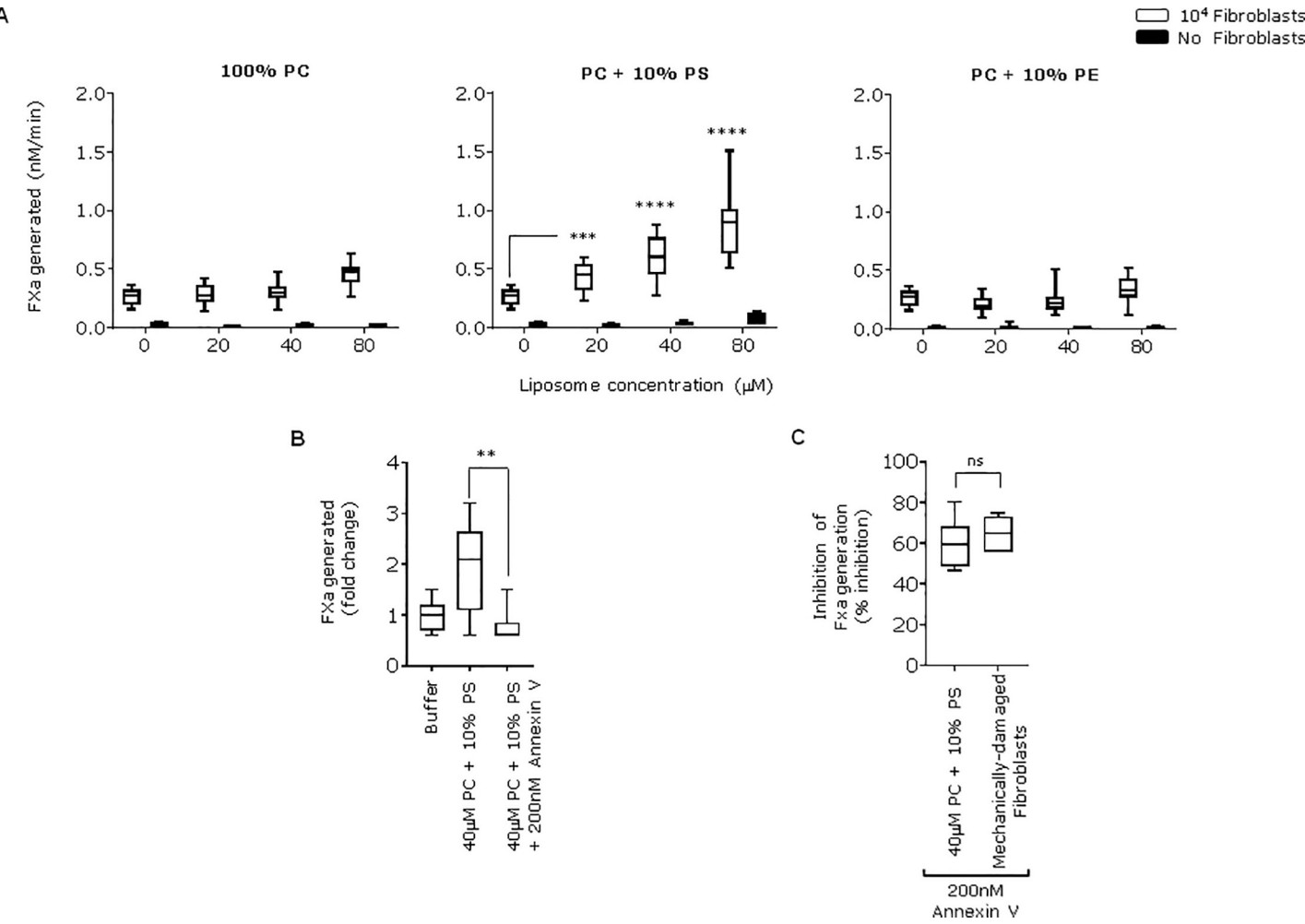

**Fig 4. Addition of TF-negative liposomes with 10% PS enhances the procoagulant activity of fibroblasts. A)** Increasing concentration of liposomes comprising 100% PC, 10% PS + 90% PC or 10% PE + 90% PC were added to $10^4$ adherent fibroblasts or in absence of cells for 10 min at 37˚C. FXa production was measured, ****p <0.0001, ***p <0.001. **B)** Changes in FXa production by adherent fibroblasts were compared when cells were incubated with buffer only or with either the PC + 10% PS liposomes or after pre-incubating these liposomes with 200 nM Annexin V for 30min at room temperature, **p <0.001. **C)** Addition of Annexin V (200 nM) for 30 min to either adherent fibroblasts treated with PC + 10% PS liposomes or mechanically-damaged fibroblasts inhibited FXa production by fibroblasts at similar levels non-significant (ns). Experiments with liposomes were done three times.

(Fig 4B and 4C). These results show that PS, but not PE, in platelet-derived EVs and liposomes can enhance FVIIa-dependent FXa generation on fibroblasts.

To investigate whether PS and PE interact directly with TF, we studied soluble TF which contains a putative lipid binding site [33, 34] but not the transmembrane domain. We observed that the extracellular domain of TF bound to both PS and PE (Fig 5A and 5B) but further studies are required to understand the mechanism of this interaction.

## Discussion

Here, we show that healthy fibroblasts support low levels of TF/FVIIa-dependent FXa generation despite the high number of TF molecules expressed on their surface. We demonstrate that this low specific activity is due to minimal amounts of externally facing PS and PE. We have characterised and quantified the molecular species of total and externally facing PS and PE for the first time on these cells and shown that these differ from those found in platelets. The

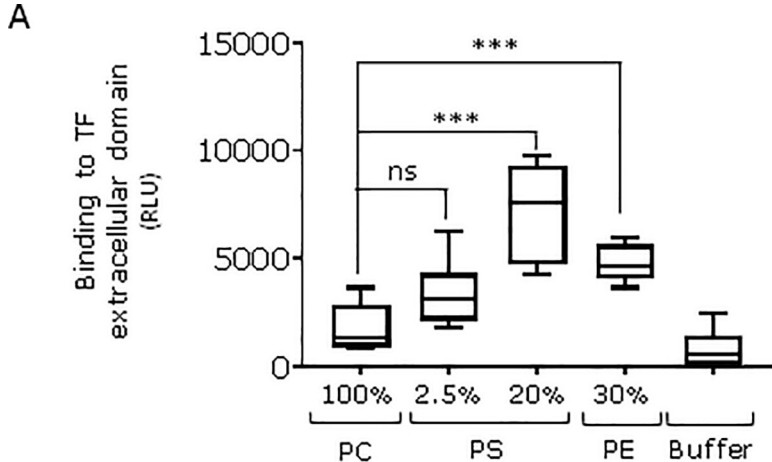

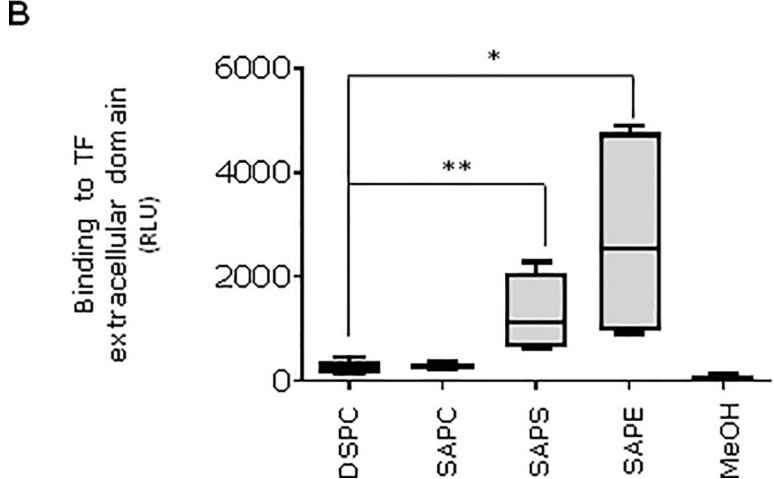

**Fig 5. TF binding to PL. A)** Liposomes with the proportion of PS or PE shown were incubated with soluble TF (see methods), ***p <0.001, non-significant (ns) P = 0.07. **B)** Pure lipids (DSPC, SAPC, SAPE, SAPS) were diluted in methanol at 3 µg/ml and air-dried for 90min at room temperature, **p <0.01, *p <0.05. Liposome or air-dried lipid ELISA procedure was performed as indicated in the methods. Graphs show relative luminescence units (RLU). These experiments were done two times.

ability of fibroblasts to support TF/FVIIa-dependent FXa generation was increased by the addition of TF-negative platelet-derived EVs and synthetic liposomes that contained PS, this effect was inhibited by annexin V. In addition, mechanical damage of fibroblasts led to increased TF/FVIIa-dependent FXa generation that was associated with, and partly dependent on, externalisation of PS and PE.

Following vessel damage, *in vivo* haemostasis requires circulating FVII/FVIIa binding to TF expressed on the surface of fibroblasts and smooth muscle cells [3]. Fibroblasts are known to express high numbers of TF molecules [6]. We confirmed this finding using a simple flow cytometry method that quantifies TF on individual cells and found that fibroblasts express 6–9 $x10^4$ molecules per cell. This TF had very limited ability to activate coagulation supporting the findings of a previous study, using a different fibroblast cell line (WI-38), that showed that about 90% of TF molecules were encrypted [6].

It has been known for many years that TF/FVIIa activity is dependent on the phospholipid composition of the membrane in which TF is expressed [35, 36]. Specifically, externally facing PS is required for TF/FVIIa cleavage of FX and this activity can be enhanced by PE [11, 12, 37]. PE in the absence of PS does not support TF activity [12]. Furthermore, regions on the cell membrane containing PS within lipid rafts are more procoagulant than regions of the same membrane containing only TF [38]. In platelets, lipid rafts play an essential role in the release of PS-exposing EVs [32]. Using annexin V binding we demonstrated that there was detectable externally-facing PS or PE on only about 2% of resting live fibroblasts. The low amount of externalised PS/PE on healthy fibroblasts was confirmed and quantified by mass spectrometry and is likely to contribute the limited activity and encryption of TF.

Physical damage to fibroblasts, simulating vessel wall injury, resulted in a 7-fold increase in TF/FVIIa-dependent FXa generation. The proportion of cells binding annexin V also increased similarly, and was associated with externalisation of large amounts of PS and PE (Fig 2). Externalisation of the most abundant PS (SOPS) increased 14-fold, while the predominant PEs (SOPE, SAPE) increased 20-fold and 14-fold, respectively (Table 1). Increased TF/FVIIa activity was suppressed by anti-TF antibody, annexin V or lactadherin. These data support the idea that increased FXa generation in mechanically disrupted fibroblasts is TF dependent and involved PS/PE. Previous studies have also show that TF is enhanced when PS is externalised, for example, during apoptosis or after freeze thawing [6]. Our data add to this understanding by quantifying the molecular species that are externalised. Lately, it has been shown that sphingomyelin, a component of the plasma membrane outer leaflet, plays a role in TF encryption and suppression of procoagulant activity without interfering with FVIIa binding [39, 40].

TF-positive EVs play key roles in thrombus formation in animal models and are incorporated into an evolving thrombus through interaction with platelet P-selectin [41, 42]. Here, we describe that TF-negative platelet-derived EVs enhance FVIIa-dependent FXa production on undamaged fibroblasts despite having virtual no ability to support FXa generation alone. The platelet-EVs and liposomes used in our experiments did not support TF activity [43] but had high levels of externalised PS and PE. The enhanced fibroblast-dependent FX activation was inhibited by annexin V, demonstrating that PS or PE were important for the effect but in this case the PS/PE were from a source that was exogenous to the undamaged fibroblasts. The absence of TF activity was confirmed when EVs/liposomes were incubated with FVIIa/FX in absence of cells and no procoagulant activity was detected. The platelet-derived EVs used in this study were either isolated from the supernatant of resting platelets or generated by calcium ionophore and EVs from both sources enhanced fibroblast TF activity. Whilst ionophore generated EVs are produced by an artificial stimulus, EVs in the supernatant of untreated platelets can be considered physiological. The effect of EVs generated by thrombin and collagen needs to be investigated. The implications of these findings for *in vivo* haemostasis is unknown because in that situation both fibroblast damage and EVs generation are likely to occur at the site of vascular injury. However, it has been shown that patients with high levels of circulating EVs did not require transfusions [44]. This will require further investigation.

Platelet-derived EVs are complex structures and to investigate whether PS or PE alone could enhance fibroblast TF activity, we used synthetic liposomes that contain no TF comprised of 10% PS, 10% PE or 100% PC. These liposomes supported virtually no FVIIa-dependent FXa generation in the absence of fibroblasts. However, the 10% PS liposomes enhanced FXa generation on fibroblasts in a concentration-dependent manner and this could be inhibited by annexin V. In contrast, liposomes that contained 10% PE had no effect. These data support the conclusion that PS, but not PE, from a source exogenous to a TF-bearing cell can enhance TF/FVIIa-dependent FXa generation. A possible explanation as to why PS but not PE containing liposomes enhanced fibroblast TF activity is that fibroblasts express virtually no

externalised PS or PE. PS is required for TF activity whilst PE can enhance the effect of PS but does not support TF activity on its own. Therefore, supplying a source of PS to fibroblast-bound TF might have an effect whilst supplying PE may not because there would still be insufficient PS available.

Here, we describe a process that enhances initiation of coagulation *in vitro* where PS-bearing EVs significantly enhance cell-bound TF activity. The potential mechanisms for this effect could include incorporation of PS into the TF-bearing membrane by fusion or another process. This would lead to an increased amount of PS in the TF-bearing membrane which is well known to enhance TF activity. If this were the mechanism then incorporation would need to be rapid because enhancement occurs within 10 minutes in our *in vitro* experiments. Alternative explanations include binding and delivering FX to TF, or by direct interaction between TF and PS enhancing TF cofactor activity through a conformational change. A direct interaction between regions of TF and PS has been suggested previously by molecular dynamics simulations that identified key amino acids which interact with the PS head group [33, 34]. These studies have suggested the presence of a putative extracellular lipid-binding region on TF, where amino acid substitutions where shown to impact directly on TF function. Here we showed that both PS and PE could bind the extracellular domain of TF [33, 34]. This suggests that direct interaction of phospholipids with TF is less likely because PE containing liposomes did not enhance TF/FVIIa function.

EVs are released from several cell types including erythrocytes, leukocytes, platelets and endothelial cells. The clinical relevance of EVs has been recognised in arterial and venous thrombosis [42], sepsis [45] and cell-based therapies [46]. Monocyte-derived and platelet-derived EVs are of similar size (100-300nm) but have distinct procoagulant activities. Specifically, monocyte EVs contain TF and independently support thrombin generation and fibrin formation while platelet EVs required the addition of TF (EVs from THP-1 cells or Innovin) to increase the rate of FXa generation [47]. Separately, it has been demonstrated that the procoagulant activity of platelet-derived EVs rely on PS exposure, and that annexin V reduces thrombin generation in a dose-dependent manner [48].

In conclusion, we report the detailed characterisation of PS and PE in a healthy TF-bearing membrane and investigate phospholipid-dependent mechanisms that can enhance TF/FVIIa-dependent FXa generation on fibroblasts *in vitro*. These observations suggest that PS added to healthy TF bearing cells may enhance cell-bound TF activity, ultimately facilitating initiation of coagulation. Further investigation is required to assess whether these mechanisms have relevance to physiological haemostasis *in vivo*.

## Supporting information

**S1 File.**
(XLSX)

## Author Contributions

**Conceptualization:** Marcela Rosas, Peter V. Jenkins, Valerie B. O'Donnell, Peter W. Collins.

**Data curation:** Marcela Rosas.

**Formal analysis:** Marcela Rosas.

**Funding acquisition:** Valerie B. O'Donnell, Peter W. Collins.

**Investigation:** Marcela Rosas, David A. Slatter, Samya G. Obaji, Jason P. Webber, Jorge Alvarez-Jarreta, Christopher P. Thomas, Maceler Aldrovandi, Victoria J. Tyrrell.

**Methodology:** Marcela Rosas, David A. Slatter, Samya G. Obaji, Jason P. Webber, Christopher P. Thomas, Maceler Aldrovandi, Victoria J. Tyrrell, Peter V. Jenkins, Valerie B. O'Donnell, Peter W. Collins.

**Project administration:** Marcela Rosas, Valerie B. O'Donnell, Peter W. Collins.

**Resources:** Marcela Rosas, Victoria J. Tyrrell, Peter W. Collins.

**Supervision:** Peter V. Jenkins, Valerie B. O'Donnell, Peter W. Collins.

**Writing – original draft:** Marcela Rosas, Peter V. Jenkins, Valerie B. O'Donnell, Peter W. Collins.

**Writing – review & editing:** Marcela Rosas, David A. Slatter, Samya G. Obaji, Jason P. Webber, Jorge Alvarez-Jarreta, Christopher P. Thomas, Maceler Aldrovandi, Victoria J. Tyrrell, Peter V. Jenkins, Valerie B. O'Donnell, Peter W. Collins.

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
