## [Decision Letter · Decision Letter 0]

17 Jul 2020

PONE-D-20-17857

The procoagulant activity of tissue factor expressed on fibroblasts is increased by tissue factor-negative extracellular vesicles

PLOS ONE

Dear Dr. Collins,

Thank you for submitting your manuscript to PLOS ONE. After careful consideration, we feel that it has merit but does not fully meet PLOS ONE’s publication criteria as it currently stands. Therefore, we invite you to submit a revised version of the manuscript that addresses the points raised during the review process.

The study has been revised by an expert in the field and myself. The study is relevant to the field and its conclussion is supported by the experimental data. The study is technically sound. It would be important to highlight the specific novel aspects provided by the study, as suggested by the reviewer.

We look forward to receiving your revised manuscript.

Kind regards,

Pablo Garcia de Frutos

Academic Editor

PLOS ONE

Journal Requirements:

2. In your Methods section, please provide additional details regarding the cell lines used in your study and ensure you have described the source. For more information regarding PLOS' policy on materials sharing and reporting, see https://journals.plos.org/plosone/s/materials-and-software-sharing#loc-sharing-materials, and for more information on PLOS ONE's guidelines for research using cell lines, see https://journals.plos.org/plosone/s/submission-guidelines#loc-cell-lines.

Reviewers' comments:

Reviewer's Responses to Questions

**Comments to the Author**

1. Is the manuscript technically sound, and do the data support the conclusions?

Reviewer #1: Yes

2. Has the statistical analysis been performed appropriately and rigorously? 

Reviewer #1: Yes

3. Have the authors made all data underlying the findings in their manuscript fully available?

Reviewer #1: Yes

4. Is the manuscript presented in an intelligible fashion and written in standard English?

Reviewer #1: Yes

5. Review Comments to the Author

Reviewer #1: The manuscript “The procoagulant activity of tissue factor expressed on fibroblasts is increased by tissue factor-negative extracellular vesicles” by Rosas and colleagues presents enhanced TF activity by externalised procoagulant phospholipids (PS) by damaged fibroblasts or as a result of membrane fusion with platelet vesicles as a novelty. This is directly the main comment: it is known for many years that PS phospholipids are essential for binding of the vitamin K-dependent proteins in a calcium dependent manner. PS exposure on the outer layer of the bilayer can be the result of apoptosis, cell activation, cell damage or membrane fusion with vesicles. Demonstrating enhanced factor Xa generation/activity by cells having both TF and PS exposure isn’t that novel. However, the authors state in their abstract: “In conclusion, despite high levels of surface TF expression, healthy fibroblasts express low levels of external-facing PS and PE limiting their ability to generate FXa. PS, either externalised by damaging the cells, or provided from external sources enhanced fibroblast TF activity. These findings reveal novel mechanisms that contribute to the initiation of coagulation and have implications for our understanding of the pro-coagulant activities EVs in normal haemostasis and disease.”

The direct questions on this paragraph are:

1: What is novel about this?

2: What implications are suggested that will impact our understanding of pro-coagulant activities of EVs?

Having said this, and realising that the above comment on novelty might sound harsh, the manuscript presents novel data that should be highlighted in the abstract and manuscript. For instance., the observation that the composition of PS differs between fibroblasts and platelet derived EVs has not been reported previously. Also, the fusion of vesicles with intact cells has been shown but not in relation to procoagulant activity and for healthy cells. One side comment on this, combining cultured fibroblasts with isolated platelet vesicles might be different from the in vivo situation and should be discussed in more detail.

Overall, despite my initial rough comment on novelty, the authors should be applaud for their extensive work on characterising the phospholipid content of cultured fibroblasts and studies on membrane fusion between the cells and isolated vesicles. Changing the focus of the novelty could improve the manuscript, thereby making it suitable for publication in PlosOne. Therefore, I suggest major revision.

6. PLOS authors have the option to publish the peer review history of their article (what does this mean?). If published, this will include your full peer review and any attached files.

Reviewer #1: No

---

## [Author Response · Author response to Decision Letter 0]

7 Sep 2020

Dear Dr Garcia de Frutos

Thank you for sending the reviewer comments for our paper entitled: “The procoagulant activity of tissue factor expressed on fibroblasts is increased by tissue factor-negative extracellular vesicles” and inviting a revised version for consideration in PLOS One.

Response to comments:

We are encouraged that the work is considered to be technically sound, that experiments were performed in a rigorous manner and that the data support the conclusions. We note than there were no comments related to experimental techniques or results. 

As requested, we have added more detail about the fibroblast cell line used in the methods section and included a reference. 

The main comment raised by the reviewer relates to a lack of novelty. This is because it has been known for many years that phosphatidylserine, expressed on the outer leaflet of a tissue factor containing membrane, is required for TF function. We agree with this statement completely but we have not claimed any of our work to be novel in this regard. 

The main novel finding that we have described is that tissue factor negative platelet-derived extracellular vesicles (EVs) or synthetic PS containing liposomes enhance TF function in an undamaged cell membrane. To our knowledge this has not been reported previously. We accept that this finding may be unsurprising given the knowledge in literature but contend that it is novel. We agree that there has been previous work which shows that TF-positive EVs can fuse with cells and enhance the initiation of coagulation but contend that to date TF-negative EVs have not been investigated. Recent findings suggesting that the extracellular domain of TF can bind directly to PS and that this enhances TF function also points to alternative roles beyond phospholipid fusion for PS.

It is possible that the enhanced TF activity described is due to the EVs or lipsomes fusing with the fibroblast membrane. We discuss this as a possible explanation for our findings and also suggest other potential mechanisms that can be explored in future studies. 

We have removed the statement that our work may have implications for the understanding of pro-coagulant activities of EVs. 

We have adjusted the abstract and manuscript to better highlight some of the other novel findings.

We agree the reviewer that the in vivo situation may differ from the effect of platelet-derived EVs on fibroblasts in vitro and have emphasised this is the discussion. 

To summarise, we suggest that the novel findings reported in this paper are:

1. Extracellular vesicles, that lack TF, enhance cell-bound TF function in healthy fibroblasts.

2. This enhancement of TF function is blocked by annexin V demonstrating that it is dependent on phosphatidylserine.

3. Synthetic liposomes that contain phosphatidylserine (but not phosphatidylethanolamine) enhance TF function in fibroblasts.

4. We agree that it is well described that phosphatidylserine, present in the cell membrane, is important for controlling TF function. However, the exact molecular composition of phospholipids in a TF-bearing cell membrane has never been reported previously. Here we report the molecular species of phosphatidylserine and phosphatidylethanolamine that are present in the fibroblast cell membrane and show that these molecular species differ from those found in platelets. 

5. In order to support TF function it is known that phosphatidylserine must be in the outer leaflet of the fibroblast membrane. Here, for the first time, we quantify the amount of each molecular species of phosphatidylserine and phosphatidylethanolamine on the external leaflet of a fibroblast membrane. Thus this is the best description to date of the phospholipid environment of TF in a cell membrane. 

6. We characterise and quantify the molecular species of phosphatidylserine and phosphatidylethanolamine on the surface of platelet derived EVs.

7. We show that unperturbed fibroblasts have very limited amounts of external facing phosphatidylserine and phosphatidylethanolamine. We accept that this has been shown previously using techniques such as annexin V staining, however, we take this further by quantifying the amounts of each molecular species in the external leaflet.

8. We show that mechanically disrupted fibroblasts have increased externally facing phosphatidylserine and phosphatidylethanolamine and we have quantified the amount of each molecular species in the external leaflet of the membrane. 

9. Mechanical disruption of the fibroblasts results in enhanced TF function which can be blocked by inhibiting phosphatidylserine and phosphatidylethanolamine. 

10. We also described an assay to show TF interaction with PLs and showed that PS and PE either on the membrane of the liposome or on their own can interact with TF.

We hope that the changes we have made to the manuscript will make it suitable for publication in PLOS One and look forward to your opinions. 

Yours sincerely

Peter Collins and Marcela Rosas

---

## [Decision Letter · Decision Letter 1]

22 Sep 2020

The procoagulant activity of tissue factor expressed on fibroblasts is increased by tissue factor-negative extracellular vesicles

PONE-D-20-17857R1

Dear Dr. Collins,

We’re pleased to inform you that your manuscript has been judged scientifically suitable for publication and will be formally accepted for publication once it meets all outstanding technical requirements.

Kind regards,

Pablo Garcia de Frutos

Academic Editor

PLOS ONE

Additional Editor Comments (optional):

Reviewers' comments:

Reviewer's Responses to Questions

**Comments to the Author**

1. If the authors have adequately addressed your comments raised in a previous round of review and you feel that this manuscript is now acceptable for publication, you may indicate that here to bypass the “Comments to the Author” section, enter your conflict of interest statement in the “Confidential to Editor” section, and submit your "Accept" recommendation.

Reviewer #1: All comments have been addressed

2. Is the manuscript technically sound, and do the data support the conclusions?

Reviewer #1: Yes

3. Has the statistical analysis been performed appropriately and rigorously? 

Reviewer #1: Yes

4. Have the authors made all data underlying the findings in their manuscript fully available?

Reviewer #1: Yes

5. Is the manuscript presented in an intelligible fashion and written in standard English?

Reviewer #1: Yes

6. Review Comments to the Author

Reviewer #1: Thank you for addressing the raised comments adequately and modifying the manuscript accordingly. Surprisingly, I have to say something in at least 100 characters. Please consider the last two sentences as space filling for to fulfil the criteria.....

7. PLOS authors have the option to publish the peer review history of their article (what does this mean?). If published, this will include your full peer review and any attached files.

Reviewer #1: No

---

## [Editor Report · Acceptance letter]

28 Sep 2020

PONE-D-20-17857R1 

The procoagulant activity of tissue factor expressed on fibroblasts is increased by tissue factor-negative extracellular vesicles 

Dear Dr. Collins:

I'm pleased to inform you that your manuscript has been deemed suitable for publication in PLOS ONE. Congratulations! Your manuscript is now with our production department. 

Kind regards, 

on behalf of

Dr. Pablo Garcia de Frutos 

Academic Editor

PLOS ONE